# Anti-Inflammatory Effects of Dietary Polyphenols through Inhibitory Activity against Metalloproteinases

**DOI:** 10.3390/molecules28145426

**Published:** 2023-07-15

**Authors:** Takuji Suzuki, Tomokazu Ohishi, Hiroki Tanabe, Noriyuki Miyoshi, Yoriyuki Nakamura

**Affiliations:** 1Department of Food Science and Nutrition, Faculty of Human Life and Science, Doshisha Women’s College of Liberal Arts, Kamigyo-ku, Kyoto 602-0893, Japan; 2Laboratory of Oncology, Institute of Microbial Chemistry (BIKAKEN), Microbial Chemistry Research Foundation, Shinagawa, Tokyo 141-0021, Japan; ohishit@bikaken.or.jp; 3Institute of Microbial Chemistry (BIKAKEN), Numazu, Microbial Chemistry Research Foundation, Numazu, Shizuoka 410-0301, Japan; 4Department of Nutritional Sciences, Faculty of Health and Welfare Science, Nayoro City University, Nayoro, Hokkaido 096-8641, Japan; htanabe@nayoro.ac.jp; 5Graduate School of Integrated Pharmaceutical and Nutritional Sciences, University of Shizuoka, Suruga-ku, Shizuoka 422-8526, Japan; 6Tea Science Center, University of Shizuoka, Suruga-ku, Shizuoka 422-8526, Japan

**Keywords:** inflammatory diseases, matrix metalloproteinases, dietary polyphenols, chlorogenic acid, curcumin, EGCG, genistein, quercetin, resveratrol, molecular docking analysis

## Abstract

Matrix metalloproteinases (MMPs) are zinc-dependent metalloproteinases that play important roles in a variety of diseases, including cancer, cardiovascular disease, diabetes, obesity, and brain diseases. Dietary polyphenols are thought to have a variety of beneficial effects on these diseases characterized by inflammation. Clinical studies have demonstrated that MMPs are in most cases upregulated in various inflammatory diseases, including osteoarthritis, rheumatoid arthritis, inflammatory bowel disease, and Alzheimer’s disease. Studies using patient-derived human samples, animal studies, and cellular experiments have suggested that polyphenols may be beneficial against inflammatory diseases by suppressing MMP gene expression and enzyme activity. One important mechanism by which polyphenols exert their activity is the downregulation of reactive oxygen species that promote MMP expression. Another important mechanism is the direct binding of polyphenols to MMPs and their inhibition of enzyme activity. Molecular docking analyses have provided a structural basis for the interaction between polyphenols and MMPs and will help to explore new polyphenol-based drugs with anti-inflammatory properties.

## 1. Introduction

Dietary polyphenols have been demonstrated to have a variety of beneficial effects on human diseases such as cancer, obesity, diabetes, cardiovascular disease, and neurodegenerative disorders [1,2,3,4,5]. Inflammation is a hallmark of such diseases [6,7,8], and polyphenols can be considered to contribute to their prevention.

Matrix metalloproteinases (MMPs), which degrade extracellular matrix proteins, are zinc-dependent metalloproteinases that play an important role in inflammation [9]. At least 23 human MMPs are known and have been implicated in a variety of diseases, including cancer, cardiovascular disease, diabetes, inflammation, and brain disease [10,11]. There is growing evidence that dietary polyphenols exert health benefits against cancer and other inflammatory diseases.

In our previous review, we discussed the anticancer property of epigallocatechin-3-O-gallate (EGCG) in terms of its inhibitory activity against MMPs [2]. Recent comprehensive review articles have also discussed the role of MMPs in rheumatoid arthritis (RA) and musculoskeletal diseases [10,11]. Our research group has focused on the health effects of dietary polyphenols such as chlorogenic acid (CGA), curcumin (CUR), EGCG, genistein (GEN), quercetin (QUE), and resveratrol (RES) (Figure 1) and published several review articles including recent ones [12,13].

This review provides updated information on effects of these dietary polyphenols on MMPs in relation to their anti-inflammatory effects with mechanistic aspects including evidence from molecular docking studies.

This review provides updated information on important dietary polyphenols, chlorogenic acid (CGA), curcumin (CUR), EGCG, genistein (GEN), quercetin (QUE), and resveratrol (RES) (Figure 1) on MMPs in relation to their anti-inflammatory effects with mechanistic aspects including evidence from molecular docking studies. 

## 2. Roles of MMPs in Human Inflammatory Diseases

Clinical studies have demonstrated that in most cases, MMPs are upregulated in various inflammatory diseases. Example studies for 13 MMPs (1, 2, 3, 7, 8, 9, 10, 11, 12, 13, 14, 25, and 26) are discussed below. 

### 2.1. Rheumatoid Arthritis and Osteoarthritis 

In 115 patients with rheumatoid arthritis (RA), 20 with osteoarthritis (OA), 28 with psoriasis arthritis, 24 with ankylosing spondylitis, 3 groups with systemic autoimmune diseases, and 30 healthy controls were examined; MMP1 levels were elevated in RA, OA, psoriasis arthritis, systemic lupus erythematosus, and mixed connective tissue disease [14]. MMP3 levels were markedly elevated in RA patients compared to controls and disease groups including OA and systemic lupus erythematosus and could be correlated to the clinical activity of RA.

A study by Mahmoud et al. [15], in which 22 patients with RA, 10 patients with OA and 10 healthy control subjects participated, found that serum MMP1 was elevated in both patients compared to controls. The levels of MMP1 and MMP3 in the synovial fluid of RA patients were significantly higher than in OA fluids. Tchetverikov et al. [16] demonstrated that serum MMP3 levels were a good biomarker of disease activity and useful in predicting joint damage progression in early RA. Similarly, in psoriatic arthritis patient serum levels of MMP3 (22.23 ± 19.25 ng/mL) were higher than those of controls (9.08 ± 3.13 ng/mL) [17]. However, the results for MMP1 appear to be inconsistent in that the serum MMP1 levels of psoriatic arthritis patients (4.40 ± 2.37 ng/mL) were significantly lower compared to the control levels (7.27 ± 3.78 ng/mL) [17].

When synovial biopsy samples were examined according to Kellgren–Lawrence grading from grade I (doubtful OA) to grade IV (severe OA), the levels of MMP1, MMP2, and MMP9 were higher in grades II to IV than that in grade I and that of MMP13 was higher in grades II to III than that in grade I [18] (Table 1). The highest levels of MMP1 and MMP13 were observed in grade II and those of MMP2 and MMP6 in grade III. The finding is consistent with the results of previous meta-analysis showing that the protein levels of MMP1, MMP2, and MMP9 were higher in patients with OA than those in controls [19]. Tchetverikov et al. [16] demonstrated that serum levels of the MMP3 were a good biomarker of disease activity and useful to predict joint damage progression in early RA.

### 2.2. Alzheimer’s Disease 

Chronic inflammation is a hallmark of Alzheimer’s disease (AD)-related neuropathological events, leading to neuronal dysfunction and death [20].

Based on a study with 31 patients with AD and 15 normal controls, Gu et al. [21] concluded that MMP9 is a good inflammatory biomarker for AD, since its levels in plasma neuronally derived extracellular vesicles were higher in AD patients than in normal controls.

A clinical study determining MMP2, MMP9, and MMP10 levels in cerebrospinal fluid (CSF) and plasma in 52 AD patients, 26 matched controls, and 24 vascular dementia patients led Duits et al. [22] to implicate the involvement of MMP2 and MMP10 in AD pathology. Sorrentino et al. [23] showed higher levels of MMP7, IL4, IL6, IL13, Chemokine C-C Motif Ligand (CCL)17, and C-X-C motif chemokine ligand (CXCL)13 in the brain biopsy samples from AD patients compared to those of control subjects.

### 2.3. Other Inflammatory Diseases

A prospective, cross-sectional, multicenter study found that serum concentrations of MMP2, MMP7, and MMP9 were higher in chronic hepatitis C patients than in healthy subjects [24]. MMP7 was a good diagnostic indicator of advanced disease stage. These MMPs may be targets of therapy against liver fibrosis in these patients.

In blood samples from patients with Buerger’s disease, an inflammatory occlusive disorder, the blood levels of MMP9, and intercellular adhesion molecule-1 (ICAM-1), and high mobility group box-1 were significantly higher than those from controls, and no differences were detected in MMP2 and MMP11 levels [25].

In hidradenitis suppurativa, a chronic inflammatory disease, higher expression of granulocyte colony-stimulating factor was observed in the skin samples from of patients [26]. The increase in the granulocyte colony-stimulating factor receptor levels was correlated with upregulation of molecules including formyl peptide receptor 1, tumor necrosis factor receptor superfamily member 10C, hematopoietic cell kinase and hexokinase 3, and MMP25. These proteins are known to contribute to prolonged activation of neutrophils by components of bacteria and damaged host cells. Membrane-bound MMP25 inactivates α1-proteinase inhibitor, which protects neutrophil-mediated proteolytic tissue damage by inhibiting neutrophil elastase, cathepsin G, and proteinase 3.

Crohn’s disease (CD) is one of the immune-mediated inflammatory bowel diseases [27]. Immunohistochemical examination of colon samples from patients with CD showed that immunosuppressive treatment resulted in an overall decrease in neutrophil MMP9 and MMP26 levels along with a decrease in stromal tissue inhibitor of metalloproteinase (TIMP)-1 and TIMP3 levels [28]. MMP1 was detected in the stroma and surface epithelium of all samples and little difference was observed between the pre-treatment and post-treatment samples. MMP7 expression was detected in the epithelium in seven out of 17 of the pre-treatment samples and two out of 17 of the post-treatment samples. MMP9 protein was generally undetectable in the surface epithelium, but stromal expression by macrophages and neutrophils was detected. MMP10 expression was observed in the surface epithelium of 16 out of 17 baseline samples and 14 out of 17 post-treatment samples. Stromal expression of MMP10 in macrophages and lymphocytes was found in all samples. MMP26 was detected in 15 of the pre-treatment biopsy samples and in seven of the post-treatment samples in endothelial cells and neutrophils in the stroma. Drug treatment reduced MMP26 expression in stromal neutrophils. No sample showed MMP26 expression in the surface epithelium.

Additional examples of studies on human subjects and clinical samples related to MMPs in inflammatory diseases are listed in V 1.

**Table 1 molecules-28-05426-t001:** Human studies on MMPs.

	Reference [PMID]	Major Findings
OA	Zeng, et al. [19]	A meta-analysis of 10 studies with 458 OA patients and 295 healthy controls indicated that the protein levels of MMP1, MMP2, and MMP9 were higher in patients with OA patients than those in the control group. Asian OA patients showed the higher protein levels of MMP1 and MMP2 compared to the controls, while Caucasians did not show such differences. In both populations, the MMP9 protein levels in OA were higher than in controls. The MMP1 and MMP9 protein levels in the synovial joint fluid were higher in OA patients than in controls. These results suggest an association between the OA pathogenesis and the increased levels of MMP1, MMP2, and MMP9 proteins.
	Sieghart et al. [29]	Interleukin (IL)1β-stimulation of primary fibroblast-like synoviocytes from OA patients activated several c-Jun N-terminal kinase (JNK) mitogen activated protein kinases (MAPKs), including extracellular signal-regulated kinase (ERK1/2), JNK, heat shock protein 27 (Hsp27), and p38 MAPK (p38). The mRNA levels of MMP2 and MMP14 were upregulated in IL1β-stimulated cells. H2S, which has anti-inflammatory properties [30], attenuated these effects as well as IL1β-induced secretion of IL6, IL8, and RANTES.
RA	Zhou, et al. [31]	Serum MMP3 levels in RA patients with moderate and severe disease activity were higher than in patients with stable RA. Levels in mild RA were not different from those in the stable RA and the healthy control group. Levels decreased in patients treated with certolizumab pegol, a tumor necrosis factor-α (TNFα) inhibitor drug for the treatment of RA. The findings suggest MMP3 as a useful serum biomarker for RA.
	Crowley et al. [32] Behera et al. [33]	Lyme disease, a form of chronic inflammatory arthritis, is transmitted by ticks (*Borrelia burgdorferi*) and patients often develop Lyme arthritis (LA). Tick infection in primary human chondrocytes caused upregulation of gene expression of MMP1, MMP3, MMP10, MMP13, MMP19, and TIMP1. Protein levels of MMP10, MMP13, and TIMP1 were increased by 72 h post infection [33]. Autoantigens in antibiotic-refractory LA were found to include MMP10 and its peptide. The levels of serum MMP10 autoantibodies in LA patients, especially antibiotic-refractory LA patients, and those of MMP10 protein in joint fluid were significantly increased. The levels of anti-MMP10 autoantibodies were correlated positively with synovial pathology, suggesting MMP10 as a useful pathological biomarker for LA.
	Yu et al. [34]	Seventy RA patients were randomly divided into two groups, and a treatment group received moxibustion, a traditional Oriental therapy with thermal stimulation by burning herbs [35], in addition to the drugs given to the control group. The decreases in the serum levels of IL1β, TNFα, MMP1, MMP3, and vascular endothelial growth factor (VEGF) in the moxibustion-treated group were more pronounced compared to the control group. The serum MMP1 and MMP3 levels are suggested to be useful biomarkers of cartilage and bone erosion in RA patients, and the efficacy of moxibustion on the clinical RA symptoms may be due to its contribution to the downregulation of MMP1 and MMP3.
	Wang et al. [36]	The expression levels of MMP3, MMP9, and MMP13 in fibroblast-like synoviocytes derived from RA patients were increased by transfection of mimic RNA of miR-145-5p, a microRNA suggested for involvement in RA development and progression. miR-145-5p inhibitor reduced these levels. The increase in MMP9 levels and the enhancement of nuclear factor (NF)-κB p65 nuclear translocation caused by miR-145-5p overexpression were attenuated by the NF-κB inhibitor, indicating the involvement of NF-κB pathway activation in MMP9 expression, suggesting that modulation of miR-145-5p is useful for the treatment of RA associated with MMPs secretion via activation of the NF-κB pathway.
Inflammatory bowel disease (IBD)	Majster et al. [37]	Analysis of inflammation-related proteins in the serum collected from IBD patients revealed a significant increase in 21 proteins including IL6 and MMP10 in the serum and a significant decrease in 4 proteins compared to the control sample. IL6 and MMP10 were also significantly increased in saliva of these patients and correlated with their expressions in the serum. These findings suggest that the oral cavity reflects ongoing intestinal inflammation, and that saliva can be used as a non-invasive source of IBD biomarkers.
	Soomro et al. [38]	In the search for a biomarker for IBD, such as CD and ulcerative colitis (UC), MMP9 and MMP12 levels in stools were found to increase significantly in both UC and CD patients. A longitudinal cohort study of 50 patients with UC showed a strong correlation of diagnostic severity of IBD with fibrinogen, MMP8, short peptidoglycan recognition protein, and TIMP2, suggesting that fecal MMP8, MMP9, and MMP12 levels are useful biomarkers for the early detection of IBD.
	Buchbender et al. [39]	mRNA expression analysis of inflammatory proteins in gingival pocket biofilms from IBD patients revealed that IL10 mRNA expression levels were higher in both CD and UC patients, and MMP7 mRNA expression levels were significantly higher in CD samples compared to the controls. In contrast, MMP7 mRNA expression levels in UC patients were not different from those in controls. The relationship between the stage of IBD progression and the mRNA expression levels of IL10 and MMP7 was not clear. These findings suggest that IL10 and MMP7 expression levels in oral biofilm may be a useful non-invasive biomarker for IBD.
	Lakatos et al. [40] Coufal et al. [41]	When serum antigen levels of MMP2, MMP7, MMP9, TIMP1, and TIMP2 were determined in 23 UC patients and 25 CD patients in comparison with 10 healthy subjects, the levels of MMP9, TIMP1, and TIMP2 were significantly higher in UC and CD, and the levels of MMP7 were higher in CD [40]. MMP9 and TIMP1 levels were positively correlated with disease activity in IBD, while MMP2 and TIMP2 levels were inversely correlated with CD activity. In contrast, lower serum levels of MMP9 and higher serum levels of MMP14 were found in patients with IBD (*n* = 85), UC (*n* = 36) and CD (*n* = 20) compared to healthy subjects (*n* = 25) [41]. A discrepancy found for MMP9 suggests the need for further studies with larger sample sizes.
AD	Boström et al. [42]	Analysis of 92 neuroinflammatory proteins in the CSF of patients with neurodegenerative diseases such as AD and frontotemporal dementia revealed that MMP10 levels were markedly increased in the CSF of both AD and mild cognitive impairment/AD patients significantly increased. In stratification by patient group and medical facility, a trend toward higher MMP10 levels was observed in all three neurodegenerative disease groups. These results suggest that elevated MMP10 levels are a common feature of AD and frontotemporal dementia, although they have different symptoms due to the different types of inflammatory proteins secreted into the CSF.
	Sorrentino, et al. [23]	Analysis of changes in the expression of pro- and/or anti-inflammatory cytokines in brain homogenate samples from AD patients and control subjects revealed increased expression of IL4, IL6, IL13, CCL17, MMP7, and CXCL13 in AD patients compared to control subjects. No significant differences were found in MMP1, MMP8, and MMP9 levels between AD patients and the control. Among the three patient clusters divided by hierarchical cluster analysis, in the class 3 group of AD, which was characterized by the low levels of amyloid β (Aβ) peptides in the brain and the longest disease duration, MMPs levels were increased. This group showed the lowest levels of almost all the molecules tested except for MMP8, MMP9, CX3CL1, and LCN2. These findings suggest that neuroinflammatory molecules such as MMP7 and CXCL13 are useful as biomarkers for the AD diagnosis.
Rhinosinusitis	Chen, et al. [43]	It is known that IL17A is markedly elevated in chronic rhinosinusitis with nasal polyps (CRSwNP). The number of IL17A-producing CD8+ T cells was increased in the CRSwNP group compared to the CR group without nasal polyps and the control group. The mRNA and protein expression levels of MMP7 and MMP9 were significantly increased in the CRSwNP group. In addition, exposure of primary human nasal epithelial cells to IL17A increased MMP9 levels. Activation of the NF-κB pathway was found to be involved in the IL17A-induced increase in MMP9 levels. These findings suggest that IL17A-induced MMP9 in the pathogenesis and tissue remodeling of CRSwNP is caused by IL17-stimulated activation of the NF-κB pathway.
	Wang, et al. [44]	Inflammatory cytokines and MMPs are known to be elevated in the tissue of CRSwNP. Sixty patients were divided into two groups to receive budesonide or placebo for 14 days. The drug treatment reduced the polyp size compared with placebo and improved symptoms. The drug reduced the expression of the pro-inflammatory cytokines IL5 and eotaxin, and increased TGFβ1 and IL10 expressions in the polyp samples. Budesonide also decreased indices of remodeling in these samples including albumin, MMP2, MMP7, MMP8, and MMP9, but increased collagen and TIMP1, TIMP2, and TIMP4 levels.
LA	Crowley et al. [32]	An immunogenic HLA-DR-presented peptide (T-cell epitope) derived from the source protein MMP10 was identified in the synovium of a patient with antibiotic-refractory LA. The level of MMP10 autoantibodies in the serum of LA patients, especially antibiotic-refractory LA patients and the amount of MMP10 protein in the joint fluid were significantly increased regardless of antibiotic-responsive or antibiotic-refractory. A positive correlation was found between anti-MMP10 autoantibodies and synovial pathology. These findings suggest MMP10 as a useful pathologic biomarker for LA.

## 3. Polyphenol’s Inhibitory Activity against MMPs in Inflammatory Diseases

### 3.1. OA and RA

Arthritis, an inflammatory joint disease, occurs through dysregulation of pro-inflammatory cytokines such as IL1β and TNFα, pro-inflammatory enzymes such as cyclooxygenase 2 (Cox-2) and lipoxygenase, expression of adhesion molecules and MMPs, and proliferation of synovial fibroblasts (SFBs) [45]. Agents that downregulate these factors are expected to prevent arthritis. Since the transcription factor NF-κB controls these factors, it is also an important target for development against arthritis-related diseases. Several dietary polyphenols, including CGA, CUR, EGCG, GEN, QUE, and RES, are known to have these arthritis-suppressing properties.

#### 3.1.1. CGA

The pathology of OA involves cartilage degradation by MMPs [46]. In an experimental model of OA in rabbits, Chen et al. found that CGA downregulated MMP1, MMP3, and MMP13 expression and upregulated TIMP1 expression at both mRNA and protein levels. CGA also attenuated IL1β-induced activation of NF-κB and degradation of inhibitor of κB (IκB)-α, accompanied by decreased cartilage degradation, suggesting its possible role in the treatment of OA. 

WIN-34B is the n-butanol fraction prepared from the two kinds of herbs for arthritis treatment in East Asian countries and contains CGA as one of the main components [47]. CGA and WIN-34B reduced the degradation of type 2 collagen by 11–62% and 13–74%, respectively, in IL1β stimulated OA cartilage explant culture. CGA caused a reduction in the culture medium levels of MMP1, MMP13, prostaglandin E2 (PGE2), and TNFα, but not MMP3 levels. CGA downregulated JNK but showed no effects on ERK. In general, WIN-34B showed better activities in cartilage-protective effects as compared to CGA. 

In an OA-like chondrocyte model in which human SW1353 chondrocytes were stimulated with IL 1β, CGA restored IL 1β-induced increases in inducible nitric oxide synthase (i-NOS)/NO, IL-6, MMP13, COX-2/PGE2, and type 2 collagen protein expression [48]. CGA inhibited the IL1β-induced inflammatory response via downregulation of the NF-κB signaling pathway, suggesting CGA as a candidate useful in the OA treatment. 

#### 3.1.2. CUR 

Microarray analysis using tips containing tissue samples of OA patients and healthy controls demonstrated that the gene expression of MMP3 in OA patients was higher compared to the controls [49]. Comparison of synovial cells derived from OA patients and healthy controls confirmed the higher protein expression of MMP3 in OA. CUR increased the expressions of fibronectin 1 and type 3 collagen, indicating decreased osteoarthritis synovial cell activity, suggesting that CUR is useful as an anti-OA agent. 

Liacini et al. [50] reported that in human femoral head OA chondrocytes, IL1β induced the activation of ERK, MAPK, and p38 upregulated MMP3 and MMP13. Like inhibitors of JNK, activator protein-1 (AP1) and NF-κB, CUR downregulated MMP3 and MMP13 in human and bovine chondrocytes, suggesting a mechanism for CUR’s inhibitory effects on these MMPs.

Stimulation with macrophage migration inhibitory factor (MIF) in SFB derived from RA patients resulted in an increase in MMP1 and MMP3 mRNA levels, along with those of c-jun, c-fos, and IL-1β [51]. Similar but weaker effects of MIF were also observed in OA SFBs. CUR, GEN, and the PKC inhibitors attenuated these effects of MIF, suggesting that MIF promotes cartilage degradation by upregulating these MMPs via pathways including tyrosine kinase, protein kinase C, and AP1.

When SW982 cells stimulated with IL1β, IL6, or TNFα were used as the RA model synoviocytes, CUR was found to reduce the viability of these cells, restore the levels of MMP1 gene expression increased by stimulation with IL1β or TNFα, and decrease TNFα protein production [52]. CUR decreased the protein level of the ribosomal protein kinase P70S6K1, leading to a reduction in the activity of the cell-surviving mTOR pathway.

In a collagen-induced RA model in rats, administration of CUR (200 mg/kg) daily for 3 weeks reduced the inflammatory features including synovial hyperplasia [53]. CUR inhibited the mTOR pathway like the inhibitor rapamycin and the infiltration of inflammatory cells into the synovium. CUR attenuated the increase in expression of IL1β, TNFα, MMP1, and MMP3 in these RA rats. 

#### 3.1.3. EGCG

In human OA chondrocytes derived from OA cartilage, stimulation with advanced glycation end-products (AGE)-modified bovine serum albumin induced gene expression and production of TNFα and MMP13, both of which were attenuated by EGCG [54]. The AGE-mediated activation and DNA binding activity of NF-κB were inhibited by EGCG via suppression of the IκBα degradation. Thus, EGCG can inhibit cartilage degradation by suppressing AGE-mediated activation of OA chondrocytes and may be useful for the treatment of inflammatory arthritis in which AGEs play a major role. 

In human OA chondrocyte CHON-001 cells, stimulation with IL1β decreased cell viability and upregulated mRNA and protein levels of miR-29b-3p and MMP13 [55]. Protein levels of IL6 were also increased. EGCG attenuated these effects. MiR-29b-3p mimics reversed these EGCG’s effects and PTEN overexpression cancelled the effects of miR-29b-3p mimics on cell viability and the levels of Bcl-2, MMP13, IL6, Bax, and cleaved caspase 3. EGCG appears useful for the treatment of OA.

In a mouse OA model that received a tissue incision, the articular cartilage of EGCG-treated mice developed less erosion and exhibited reduced staining for MMP13 and a disintegrin and metalloproteinase with thrombospondin motifs 5 (Adamts-5) compared to vehicle-treated controls [56]. EGCG caused reduced levels of mRNA expression of MMP1, MMP3, MMP8, MMP13, Adamts-5, IL1β, and TNFα in cartilages. EGCG did not affect MMP gene expression. 

Excessive osteoclast activation is one of the common features in many bone-loss diseases such as osteoporosis and RA [57]. In cultures of rat osteoclast precursor cells and mature osteoclasts, EGCG decreased the number of multinucleated osteoclasts and actin rings as well as the activities of MMP2 and MMP9 compared to the control culture [58]. Inhibition of osteoclast formation and differentiation was also observed. 

Joint cartilage and bone destruction in RA patients is caused by MMPs produced by RA SFBs and TNFα contributes significantly to their production. In these cells, EGCG suppressed TNFα-induced production of MMP1 and MMP3 at the protein and mRNA levels with concomitant downregulation of ERK1/2, p38, and JNK [59]. EGCG induced the inhibition of the binding of AP1 to its response elements in the SFB.

In RA, plasma levels of IL1β are elevated compared to those of normal subjects [60], IL1β-stimulated primary human SFBs derived from RA patients showed increases in the production of MMP2. EGCG and EGC inhibited MMP2, IL6, and the IL8 production in these cells [61]. EC had no effect. EGCG inhibited p38 activation and abolished NF-κB and p-c-Jun nuclear translocation in response to IL1β activation.

In RA synovium-derived fibroblasts, EGCG at 10-20 μM inhibited the production of IL1β-induced epithelial neutrophil activating peptide-78, RANTES, and CXCL1, but monocyte chemotactic protein 1 (MCP1) production was inhibited at much higher concentrations [62]. EGCG inhibited constitutive, IL1β-induced, and chemokine-mediated MMP2 activity. EGCG inhibited the phosphorylation of PKCδ and the activation and nuclear translocation of NF-κB in these cells stimulated with IL1β.

EGCG inhibited IL1β-induced IL6 production and trans-signaling in RA SFBs by enhancing soluble gp130 production [63]. In a rat model, EGCG ameliorated adjuvant-induced arthritis by reducing the IL6 levels in serum and joints. EGCG reduced the activity of MMP2 induced by IL6/soluble IL6 receptor complex through increasing the soluble gp130 synthesis, which led to the reduction in MMP2 activity in RA SFBs. The results suggest that EGCG-induced reduction in MMP2 activity in the joints may be due to reduction in circulating and joint IL6 concentrations and increased synthesis of soluble gp130. 

#### 3.1.4. GEN

Upregulation of IL1β was reported in the synovial fluid and cartilage of patients with OA [64]. In chondrocytes derived from osteoarthritis patients, GEN suppressed the IL1β-induced expression of NOS2, COX-2, MMP1, MMP2, MMP3, and MMP13 [65]. GEN reduced reactive oxygen species (ROS) generation in IL1β-stimulated OA chondrocytes and attenuated the IL1β-induced decrease in Nrf2/hem oxidase-1. Knockdown using si-Nrf2 caused inhibition of heme oxidase-1 expression and induced ROS generation in these cells stimulated with IL1β. 

In rheumatoid synoviocytes derived from patients, IL1β or TNFα upregulated expression of MMP9 and MMP2 [66]. Epidermal growth factor (EGF) enhanced the expression of MMP9 but not of MMP2; GEN decreased the production of MMP9 more efficiently than that of MMP2 in the cells induced by IL1β or TNFα; GEN and SP600125, the JNK inhibitor, decreased IL1β, TNFα, or EGF-induced MMP9 expression. 

In a model of knee osteoarthritis in rats, intra-articular injections of iodoacetate increased the expression of MMP8, MMP13, and Indian hedgehog but decreased the expression of type 2 collagen, Sox5, and Sox6 [67]. GEN partially attenuated these effects. The number of chondrocytes was significantly higher in the articular cartilage of the GEN-treated group compared to the untreated group. These findings suggest an osteoprotective effect of GEN. 

#### 3.1.5. QUE

OA is characterized by features such as cartilage matrix degradation, the production of inflammatory cytokines, apoptosis of chondrocyte and activation of macrophages in the synovial fluid [68]. In an OA model of rats with a single intra-articular injection of monosodium iodoacetate, the protein levels of MMP3, MMP13, Adamts-4, and Adamts-5 were upregulated compared to the control rats [69]. QUE attenuated the upregulation of these protein levels and restored the protein levels of type 2 collagen and aggrecan in the cartilage as well. QUE also reduced the upregulated levels of several inflammatory cytokines including IL1β, IL6, and IL10 and growth factors such as VEGF in the synovial fluid and/or serum of OA rats. 

QUE inhibited unstimulated and IL1β-stimulated cell proliferation of rheumatoid SFBs, and also the mRNA and protein expression of MMP1, MMP3, and COX-2 [70]. QUE inhibited the phosphorylation of ERK1/2, p38, JNK and the activation of NF-κB upregulated by IL1β. QUE may be useful for the management of RA via the inhibition of SFB proliferation and MMP-mediated joint destruction. 

#### 3.1.6. RES

IL1β stimulation of chondrocytes from mouse condylar cartilage induced apoptosis and upregulated the mRNA expression of COX-2, p65, MMP1, and MMP13 [71]. RES attenuated these effects. In a collagenase-induced temporomandibular joint OA model, mRNA expression of COX-2, p65, MMP1, and MMP13 were upregulated in condylar cartilage compared to normal controls. RES injection prevented these collagenase-induced changes. 

IL1β stimulation of chondrocytes from OA patients increased the mRNA levels of MMP1, MMP3, and MMP13 and protein expression levels of enzymes including COX-2, i-NOS, MMP1, MMP3, and MMP13 compared to the unstimulated cells [72]. These effects were cancelled by RES. RES restored the reduced protein expression of type 2 collagen as well. In addition, RES restored the phosphorylation levels of p65 and IκB upregulated by IL1β, indicating that RES’s activity is associated with its downregulation of NF-κB signaling pathway. 

Stimulation with TNFα in chondrocytes from OA patients resulted in upregulation of COX-2, MMP1, MMP3, MMP13, and PGE2 production and MMP2 and MMP9 activity [73]. RES attenuated the TNFα’s effects and knockdown of silencing information regulator 2-related enzyme 1 (Sirt1) by siRNA reduced RES-mediated anti-inflammatory activity. RES attenuated the upregulated protein expression of p65 caused by TNFα, suggesting its downregulation of the NF-κB pathway. 

Elayyan et al. [74] demonstrated that Sirt1 downregulated MMP13 in human OA chondrocytes via downregulation of the transcription factor LEF1, which is a cofactor of β-catenin in transcription. RES also downregulated gene expression and enzyme activity of MMP13 which can be explained by RES’s ability to downregulate LEF1 via upregulation of Sirt1 [75]. 

RES inhibited spontaneous PGE2 production in cultured cartilage explants from OA patients and abrogated PGE2 upregulation in IL1β-stimulated chondrocytes [76]. The results were consistent with the findings on the effects of RES on gene and protein expression of COX-2. In addition, RES prevented downregulation of proteoglycan synthesis and upregulation of MMP1, MMP3, and MMP13 caused by IL1β in OA cartilage. In unstimulated OA cartilage, RES increased proteoglycan synthesis and downregulated MMP13. 

The Sirt1 protein and mRNA levels in fibroblast-like synoviocytes isolated from RA synovial tissues were lower compared to normal fibroblast-like synoviocytes [77]. RES decreased the expression of MMP1 and MMP13 and suppressed the invasive ability of these cells. In a collagen induced-arthritis rat model, an intraperitoneal injection of RES prevented joint damage and downregulated MMP1 and MMP13 expression in synovial tissues. 

In IL1β-stimulated human chondrocytes derived from healthy femoral head articular cartilage, CUR and RES synergistically downregulated NF-κB-regulated gene products related to inflammation such as COX-2, MMP3, and MMP9 and apoptosis such as Bcl-2 and Bcl-xL along with suppression of IL1β-mediated NF-κB activation [78]. These two polyphenols similarly inhibited nuclear translocation of activated NF-κB, but differed in that CUR but not RES inhibited the phosphorylation of IκBα. 

In chondrocytes isolated from pig joints, RES inhibited AGE-induced expression of i-NOS and COX-2 and production of NO and PGE2 via inhibition of the IKK-IκBα-NF-κB and JNK/ERK/AP1 signaling pathways upregulated by AGE [79]. RES also suppressed AGE-stimulated gene expression and activity of MMP13 and prevented AGE-mediated degradation of type 2 collagen. 

### 3.2. Asthma

In a mouse model of chronic asthma, sensitization with ovalbumin (OVA) induced inflammation dominated by eosinophils, leading to fibrosis and airway remodeling [80]. CUR administered through an intranasal route prevented airway inflammation and pulmonary fibrosis along with the downregulation of MMP9 activity in bronchoalveolar lavage fluid and lung tissue. 

Bronchoalveolar lavage fluid of OVA-induced asthmatic mice exhibited enhanced activities of MMP2 and MMP9 [81]. Intranasal CUR reduced protein expressions of MMP9, histone deacetylase 1, histone H3 acetylation at lysine 9, and NF-κB p65, suggesting that intranasal administration of CUR can be effective in preventing asthma severity. 

In the lungs of asthmatic model mice exposed to OVA as an antigen and the endotoxin lipopolysaccharide (LPS) simultaneously, airway inflammation, enhanced production of ROS, increased mRNA expression of MMP9, TIMP1, TGFβ1, IL13, type 1 collagen and TLR-4, and activation of MAPK pathway enzymes were observed [82]. The expression of proteins involved in signal transduction such as MAPKs (p-ERK, p-JNK, and p-p38) and TLR-4, and enzymes including COX-2 and lipoxygenase-5 were also detected. CUR attenuated these changes. 

#### EGCG

In lung tissues from the asthma model mice, mucus production and gene expression of mucin 5B, p38, and MMP9 were elevated, compared to the non-asthmatic mice [83]. EGCG attenuated these changes in the asthmatic mice. In the nasal epithelial cells derived from allergic rhinitis patients, EGCG abrogated the upregulation of mucin 5B and MMP9 expression induced by phorbol 12-myristate 13-acetate. 

In a mouse asthma model that received intranasal administration of toluene diisocyanate, EGCG reduced the total cell number and cell counts of inflammatory cells such as neutrophils, eosinophils, and macrophages as well as ROS and TNFα levels in bronchoalveolar lavage fluid, compared to vehicle-treated control mice [84]. In the lung tissue, EGCG abrogated MMP9 protein and gene expression increased by toluene diisocyanate inhalation. EGCG also attenuated the increased expression of MMP9 protein in the lung tissues and the increased levels of MMP9 gelatinolytic activity.

### 3.3. AD

Studies have demonstrated that CGA, CUR, and EGCG may have beneficial effects on neuroprotection and cognitive function in AD [85,86,87,88] through their activities to decrease β-amyloid plaques, delay degradation of neurons, prevent inflammation, and decrease microglia formation. However, there has been no convincing evidence to indicate the involvement of MMP modulation by these dietary polyphenols in their beneficial effects on AD. 

#### 3.3.1. GEN

A 105-amino-acid carboxyl-terminal fragment (CT105) of amyloid precursor protein increased TNFα and MMP9 production in interferon-γ-treated human monocytic THP-1 cells [89]. The inhibition of TNFα by its antibodies abolished MMP9 production, suggesting its role in MMP9 production. GEN reduced both TNFα secretion and MMP9 release in the cells treated with CT105 or β-amyloid. Specific inhibitors of ERK and p38 suppressed CT105-induced effects, whereas only ERK inhibitor attenuated β-amyloid’s effects. The findings suggest that both tyrosine kinase and MAPK signaling pathways can be potential therapeutic targets for AD. 

MIF stimulation of synovial RA fibroblasts upregulated gene expression of MMP1 and MMP3 [51] and GEN abrogated the effects of MIF stimulation.

#### 3.3.2. QUE

Cholesterol oxidation products, oxysterols, have been implicated in the pathology of AD [90]. In human neuroblastoma SH-SY5Y cells, oxysterols upregulated the gene expression of inflammation-related proteins such as IL8, MCP1, and MMP9 [20]. QUE loaded in nanoparticles attenuated these effects of oxysterols, suggesting that QUE can be a new therapeutic strategy for AD via this drug delivery system.

#### 3.3.3. RES

A retrospective study on mild-to-moderate AD patients found that encapsulated RES administered orally for 52 weeks reduced the CSF levels of MMP9, but not those of MMP2, MMP3, and MMP10, compared to the placebo group [91]. The RES-induced decrease MMP9 in CSF suggests that RES may reduce central nervous system (CNS) permeability and limit the infiltration of inflammatory agents such as leukocytes into the brain, since MMP9-mediated breakdown of the basal lamina of the neurovascular gap junctions leads to increased CNS permeability. Although the placebo group showed cognitive decline as indicated by Mini-Mental Score Examination scores, no decline was observed in the RES group, suggesting that RES may contribute to cognitive improvement in AD patients.

In an animal model of AD, RES reduced Aβ42 in the hippocampus compared to the control rats; downregulated AGE receptor, MMP9, and NF-κB; and upregulated Claudin-5 [92]. These findings suggest that RES is useful for protecting against Aβ42-induced neuroinflammation via the downregulation of NF-κB and maintaining the integrity of the blood–brain barrier via the upregulation of Claudin-5 and downregulation of AGE receptor and MMP9. 

### 3.4. Other Inflammatory Diseases

Some selected studies on other inflammatory diseases in relation to the polyphenol’s action are as follows. 

#### 3.4.1. CUR

Gastric ulcer is characterized by inflammation, irritation, or erosion in the mucosal lining of the stomach [93]. In indomethacin-induced gastric ulcer in rats, ulcerated stomach extracts had an upregulation of 92 kDa pro-MMP9 activity and a moderate reduction in MMP2 activity [94]. CUR showed antiulcer activity in acute ulcers by preventing glutathione depletion, lipid peroxidation, and protein oxidation. CUR prevented gastric ulceration and promoted the healing process by inhibiting MMP9 activity and enhancing MMP2 activity. The opposite expression patterns of MMP2 and MMP9 may be due to the difference in transcriptional regulation: MMP9 but not MMP2 promoters contain several putative AP1 and NF-κB binding sites. 

In rat IBD induced by 2,4,6-trinitrobenzene sulfonic acid, Motawi et al. [95] found an increase in serum levels of TNFα and nitric oxide and in the colonic levels of MMP1, MMP3, TIMP1, and nitric oxide. The levels of these parameters were reduced by CUR.

In CUR-treated mucosal biopsies from children and adults with IBD, p38 activation and IL1β were downregulated, while IL10 was upregulated [96]. In the culture of IBD colonic myofibroblasts, CUR downregulated MMP3.

Previous data showed that intestinal subepithelial myofibroblasts in patients with CD had an increased oxidative status [97]. In culture of these cells, MMP3 was upregulated due to the increased oxidative state, since N-acetylcysteine caused a reduction in MMP3 production. In TNFα-stimulated human colonic myofibroblast 18Co cells, CUR exhibited effects similar to N-acetylcysteine. 

#### 3.4.2. EGCG

LPS promotes the inflammatory reaction of hepatitis. LPS-stimulation of L02 hepatocytes upregulated inflammation-related proteins including MCP1, TNFα, ICAM-1, VEGF, and MMP2 [98]. EGCG attenuated these effects caused by LPS-stimulation and downregulated NF-κB and MAPK signaling pathways via downregulation of p-IκBα, p65, p-p65, p-p38, p-ERK1/2, and p-AKT. 

#### 3.4.3. QUE

Human retinal pigment epithelial cells play a critical role in various retinal inflammatory diseases. TNFα increased the mRNA and protein expression of ICAM-1 along with elevation of mRNA and enzyme activity of MMP9 in these cells [99]. QUE attenuated these effects. QUE reduced the TNFα-induced phosphorylation of PKCδ, JNK1/2, ERK1/2, c-Jun phosphorylation, and AP1 promoter activity. QUR abrogated TNFα-induced NF-κB p65 phosphorylation and translocation. These findings indicate that the QUE’s downregulation of MMP9 is mediated via the downregulation of MEK1/2-ERK1/2 and PKCδ-JNK1/2-c-Jun or NF-κB signaling pathways.

#### 3.4.4. RES

IL1β stimulation decreased the content of proteoglycans in cultured adult human articular cartilage, suggesting the elevated degradation of extracellular matrix, and RES reversed the effect. IL1β stimulation of human articular chondrocytes resulted in upregulation of gene and protein expression of MMP13 and RES attenuated these effects [100]. RES inhibited the DNA-protein interaction of transcription factors activated by IL1β, including p53, AP1, NF-κB, and Sp1.

## 4. Mechanistic Considerations on Polyphenol’s Inhibition of MMPs

One important mechanism by which polyphenols exert their anti-MMP activities is likely to be via the downregulation of ROS that promote MMP expression. Another is their direct binding to MMPs, leading to inhibition of enzyme activity.

### 4.1. Inhibitory Effect of Polyphenols on MMPs via ROS-Mediated Signaling Pathways

In our previous discussion on the molecular mechanisms of EGCG for its anti-MMPs, the effects on ROS-mediated signaling were highlighted. Since CGA, CUR, GEN, QUE, and RES have similar effects on this signaling as discussed previously, the EGCG-mechanism can be applied to these polyphenols [2] and Figure 2 shows a brief depiction. The expression of MMPs is controlled by several transcription factors including NF-κB, AP1, Sp1, β-catenin, ERK1/2, p38, and Hsp27, and previous studies have shown that the activities and expression of most of these transcription factors are downregulated by these dietary polyphenols, such as EGCG (Table 2). Thus, one of the important anti-MMP mechanisms of these polyphenols is considered to be downregulation of these transcription factors which can be triggered by their antioxidative action.

Another important mechanism for MMP inhibition is likely to involve the direct binding of these polyphenols to MMP proteins. In an early stage of polyphenol’s binding to MMPs, affinity chromatography using EGCG immobilized on agarose was used [129]. The results revealed the direct binding of cancer-cell-derived MMP2 and MMP9 to EGCG. Surface plasmon resonance analysis revealed the direct interaction of catechins with MMP14 and MMP2, where catechins without a galloyl group interacted weakly as compared to those with a galloyl group [130]. Direct binding of MMP1 to QUE was demonstrated in a pull-down assay of B16 melanoma cell lysates using Sepharose beads coupled with 7-O-aminoethylquercetin [131]. 

Computational molecular docking analysis (MDA) has provided the structural basis for the interactions between polyphenols and MMPs. Recently, MDA has been applied to reveal the molecular interaction between MMPs and polyphenols.

### 4.2. MDA of Interactions between Polyphenols and MMPs

#### 4.2.1. Interaction between MMP1 and Polyphenols

##### CUR, EGCG, QUE, and RES

The MDA conducted by Priya et al. [132] showed that MMP1 interacts with CUR via hydrogen bonding involving Glu 209, Tyr 210, Glu 219, Ser239 and π–π stacking via His218, Tyr240; EGCG via Asn180, Leu181, Ala182, Leu235, Pro238, Tyr240 (hydrogen bonding), and Arg214 (π–cation bonding); QUE: Gly179, Asn180, Glu 209, Glu 219 (hydrogen bonding), and Tyr 210 and His218 (π–π stacking) (Figure 3). In addition, MMP1 interacts with GEN via Leu 181, Ala 182, Tyr 210, Glu 219 (hydrogen bonding), and His 218 (π–π stacking), and also with RES via Asn180, Ala 182, Glu 209, Glu 219 by hydrogen bonding [132].

Other studies found similar interactions for QUE–MMP1 binding. An MDA conducted by Xiao et al. [133] showed that QUE is retained in the pocket surrounded by Asn180 to Tyr240 in MMP1. The OH group on C7 in QUE forms a hydrogen bond with Glu219 in MMP1. MDA conducted by Zhang et al. [134] showed two hydrogen bonds between 4′OH and 5C in QUE and Glu219 (Figure 4). 

The binding energy was calculated to be −7.15/mol. These findings indicate that these dietary polyphenols have similar binding mode with MMP1 as exemplified by the interaction of CUR, GEN, QUE, and RES with Glu 219, and that of EGCG, QUE, and RES with Asn180.

#### 4.2.2. Interaction between MMP2 and Polyphenols

##### CGA

MDA studied by D’Abadia et al. [135] revealed that CGA is retained in the catalytic pocket composed of amino acid residues of Leu163 to Tyr223 in MMP2 (Figure 5). One of the OH groups on the dihydroxyphenyl structure in CGA forms a hydrogen bond with Glu202. Each of the two OH groups on the carboxyl group in CGA forms a hydrogen bond with Glu210. In addition, CGA shows van der Waals interactions with Leu163, Leu164, Ala165, His166, Ala167, Val198, His201, His205, His211, Pro221, and Tyr223. The minimum binding energy was not determined. 

##### CUR 

Based on the crystallographic structure of MMP2, Hosseini et al. [136] assumed that the catalytic pocket of MMP2 is composed of amino acid residues including His70, Gly81, Leu82, Leu83, His85, Ala88, Asp100, Leu116, Val117, His120, Glu121, His124, Glu129, His130, Leu137, Pro140, Ile141, Tyr142, and Try143. MDA indicated that CUR interacts with His70 and Ala86 in MMP2 through hydrogen bonding (Figure 6). Dendrosome CUR (DenCUR), a nanocarrier form of CUR derivative with oleoyl and polyethylene glycol portions [137], exhibited the greater effect on cell death compared to CUR. MDA showed that MMP2 forms hydrogen bonds with DenCUR via His70, Ala88, Asp100, and Glu129, which can explain its higher biological activity compared to CUR.

MDA conducted by Ahmad et al. [138] showed that CUR and its difluorinated benzylidene derivative bind to MMP2 through the enzyme pocket containing the hydrophobic residues Phe148, Phe115, Thr145, and Leu150 (Figure 7). One of the hydroxyl groups of CUR is involved in hydrogen bonding with Arg149 and the methoxy group on the other aryl ring interacts with Zn atom. The minimum binding energy is −7.35 kcal/mol. The CUR derivative interacts with Leu83, Ala84, and His120 through hydrogen bonding with a minimum binding energy of -6.39 kcal/mol. The inhibitory activity against MMP2 of this derivative with higher bioavailability was stronger than that of CUR. These findings may be useful to develop an effective inhibitor of MMP2 which is involved in various diseases.

##### EGCG

MDA of the active MMP2-EGCG complex revealed binding cavity of 10 A˚ around the active site residue Glu404 and that Ala192, Leu399, His403, Glu404, and Met421 are located close to the EGCG binding site [2,139]. The 15 amino acid residues (Trp151, Leu190, Leu191, Ala192, His193, Phe207, Trp213, Leu397, Leu399, Val400, His403, Glu404, Met421, Tyr425, and Tyr427) have an interaction energy of >2 kcal/mol. The A-ring in EGCG has π-σ interaction with Met421, 7OH interacts by hydrogen bonding with Leu399 and His403, 3′OH in B-ring with Glu404, and 5′OH with Ala192. The binding free energy was calculated to be −32.72 kcal/mol.

Comparison of epigallocatechin (EGC) without galloyl group and EGCG showed that EGC has no discernible inhibitory activity against MMP2 [139]. MDA shows that EGC does not interact with Trp151, Phe207, Leu397, and Tyr427, which are involved in the interaction with EGCG but interacts with Glu412, His413, and Pro423, which are not found in the interaction of EGCG with MMP2. These differences may be related to different factors, such as the number and position of hydroxyl groups affecting the conformation of MMP2, which may be responsible for the difference in activity between EGCG and EGC.

##### QUE

For the interaction between QUE and MMP2, Li et al. [140] reported that QUE is retained in the pocket surrounded by Gly80 to Tyr142 in MMP2. Leu82, Ala139, and Ile141 in MMP2 interact through a hydrogen bond and Gly80, Leu81, Leu116, Val117, His120, Leu137, Pro140, and Tyr142 are involved in hydrophobic interactions with QUE (Figure 8). The binding energy was calculated to be −8.17 kcal/mol. Similarly, MDA conducted by Erusappan et al. [141] revealed hydrogen bonding of QUE to Leu83 and Glu121 in MMP2 (Figure 9). In addition, similar to the above findings [140], the interaction with Leu137, Ala139, and Ile141 is formed via hydrogen bonding. The binding energy was calculated to be −10.1 kcal/mol in this study.

Xu et al. [142] reported that QUE binds Tyr3, Ile141, Thr143, Thr145, Asn147, and Phe148 of MMP2 (Figure 10) with an affinity of −7.90 kcal/mol. These three results are similar but not identical.

The results found by Pandey et al. [143] were somewhat different from the above three studies. These authors found that the binding energy between QUE and MMP2 is −9.11 kcal/mol with Ki value of 210.76 nM and that Leu164, Ala165, Ala217, and Ala220 in MMP2 are involved in hydrogen bonding (Figure 11). These interactions are rather similar to those in the aforementioned CGA–MMP2 complex, where CGA is located in the catalytic pocket of MMP2 consisting of amino acid residues from Leu163 to Tyr223 and interacts with amino acid residues including Leu164 and Ala165 [135].

#### 4.2.3. Interaction between MMP3 and Polyphenols

##### CUR

Docking analysis revealed that the free energy of binding of the CUR with MMP3 was −10.2 kcal/mol [144] (Figure 12). The estimated Ki value was 3.6 × 10^−8^ M. The two phenyl rings of CUR show π–π interaction with His201 on one side and with His224 and Tyr223 on the other side of the active site pocket. One enol-OH of the CUR midsection interacts with the backbone O of Leu218 by hydrogen bonding. Hydrogen bonds are also formed between phenolic-OH of CUR and backbone N of Leu164 and N of Ala165.

##### QUE

Zhang et al. [134] found that QUE, via the B-ring and 7OH, forms a hydrogen–π conjugation with Leu164 and Tyr155, respectively, in MMP3 (Figure 13). Interaction between the carbonyl group on C4 of QUE and Zn can also be observed, but no hydrogen bonding was detected. The binding energy was calculated to be −7.25 kcal/mol.

#### 4.2.4. Interaction between MMP9 and Polyphenols

##### CUR

MDA performed by Zhong et al. [145] showed that CUR forms hydrogen bonds with MMP9 via Leu188 and Ala191. This finding is in accordance with the result of a previous study showing that the catalytic cavity of MMP9 involves Leu188 [143].

##### EGCG

In the MMP9–EGCG complex, amino acid residues with an interaction energy of >5 kcal/mol are Leu187, Leu188, His401, Glu402, and Pro421, and those with an interaction energy > 2 kcal/mol are Ala189, His190, Ala191, Phe192, Val398, Ala399, His405, His411, and Met422 [146]. Among them, Leu188, Ala189, Glu402, His405, and Pro421 interact with 5′OH, 5′OH, 5″OH, 5OH, and OH in the enol form of the galloyl group, respectively, via hydrogen bonding. His401 interacts with D-ring in EGCG via π–π bonding. 

In contrast to EGCG, EGC showed no significant inhibitory activity against MMP9 [139]. MDA revealed that the key amino acid residues for EGC and MMP9 interaction are different from the EGCG–MMP interaction, with EGC having 10 and EGCG 14 interacting residues, with only nine residues in common. EGC has one additional residue, Phe181, with which EGCG interacts very little, while five additional residues, Ala189, Val398, Ala399, His401, and Met422, are also involved in the case of EGCG. Thus, these differences may be responsible for the differences in MMP9 inhibitory activity observed between EGCG and EGC.

##### QUE

MDA showed that the binding energy between QUE and MMP9 is −8.82 kcal/mol with a Ki value of 343.46 nM [143]. The catalytic cavity of MMP9 was formed by Leu 188, Leu 397, Val 398, His 401, and Leu 418, and the Met 422-Tyr 423 main chains, and QUE interacts with the active site residue Pro 415 via hydrogen bonding (Figure 14).

Yu et al. [147] found that Leu188, Ala189, Glu227, and Tyr245 in MMP9 are involved in hydrogen bonding with QUE and Glu186, Leu187, Leu222, Val223, His226, Leu243, Pro246, Met247, and Tyr248 are involved in hydrophobic interactions (Figure 15). The binding energy of MMP9 with QUE was calculated to be -10.8 kcal/mol. Similarly, Zhang et al. [134] reported that Leu188 in MMP9 interacts with QUE C-ring through hydrogen–π bonding and Glu402 interacts with 3OH through hydrogen bonding (Figure 16). The binding energy was calculated to be −6.16 kcal/mol. 

MDA and a molecular dynamics simulation lead Saragusti et al. [148] to present two models of MMP9–QUE complexes, A and B, as the most favorable structures based on factors including binding free energies. In the A model, the chromone moiety in QUE has the hydrophobic interactions between the side chain of Leu188 and Val398, and 5OH in QUE has two strong hydrogen bonds with Leu188 and Ala189 in MMP9 (Figure 17). In the B model, the benzene ring has hydrophobic interactions with Leu187, Leu188, and Val398. Additionally, the chromone ring interacts with Tyr393 and Tyr423. Two strong hydrogen bonds are formed between 3′OH and Leu188 and between 4′OH and active site Glu402, and one hydrogen bond is formed between 3 OH and the carbonyl of Pro421 (Figure 17).

Huynh et al. [149] found that QUE interacts with MMP9 by binding within the active site pocket and interacting with Leu188, Ala189, Glu 227, and Met247. The binding affinity predicted by molecular docking was −9.9 kcal/mol. Thus, all data indicate the involvement of Leu188 in the catalytic cavity of MMP9 [143].

#### 4.2.5. Interaction between MMP14 and Polyphenols

##### EGCG

MDA showed that EGCG binds to MMP14 with binding free energy of −57.61 kcal/mol [139]. The amino acid residues with interaction energy of >2 kcal/mol are Gly197, Leu199, Phe234, Leu235, Val236, Val238, His239, Glu240, His243, Ser250, Ile256, Met257, Ala258, Pro259, Phe260, Try261, Gln262, and Met264, of which Leu199, Phe234, His239, Glu240, Met257, and Gln262 are involved in hydrogen bonding [2,139]. His239 interacts with both B-ring and C-ring in EGCG via π bonding.

In contrast to EGCG, EGC did not show noticeable inhibitory activity against MMP14 [139]. MDA shows that EGC interacts with 11 amino acid residues, whereas EGCC interacts with 18 residues. Among them, only six residues are shared in the EGCG and EGC interaction with MMP14. These differences appear to contribute to the different activities of EGCG and EGC.

Table 3 shows a summary of binding energies in complexes of polyphenols and MMPs and amino acid residues involved in hydrogen bonding.

## 5. Future Perspectives

As shown above, several clinical studies have demonstrated that MMPs are useful as biomarkers for disease severity in serum and other tissue specimens. However, there are some reports that require further studies for confirmation. For example, in psoriatic arthritis patients, the serum MMP1 levels were significantly lower than the control levels [17]. MMP7 mRNA expression levels in UC patients were not different from those of controls [39]. Future studies may provide evidence to show how the genetic background is related to MMP expression in view of a finding that Asian OA patients showed higher protein levels of MMP1 and MMP2 compared to the controls, while Caucasians did not show such differences [19].

Most of the basic studies have demonstrated the inhibitory activity of polyphenols against MMPs in inflammatory diseases. However, it is not clear whether polyphenols are useful for prevention and/or therapy and this issue awaits future interventional studies.

As for mechanistic aspects, the anti-inflammatory activity of polyphenols can be explained by the antioxidant property. In the ROS-mediated signaling pathways, various transcription factors are downregulated by the action of polyphenols as antioxidants, which leads to the downregulation of gene expressions of MMPs. However, in some cases, the downregulation of transcription factors by polyphenols is not necessarily related to the downregulation of MMP expression. For example, Vicentini et al. [150] reported that QUE inhibited UV irradiation-induced NF-κB DNA binding activation in primary human keratinocytes, but no effects on MMP1 and MMP3 expression were observed, although QUE attenuated UV irradiation-induced upregulated mRNA expression of IL1β, IL6, IL8, and TNFα. Further studies are required to know whether the difference is due to the cell type used or not.

As mentioned above, miR-145-5p regulates MMP expression in RA, impacting the NF-κB pathway and suggesting its potential role in targeted RA therapies [36]. Our previous report suggested that CGA, CUR, EGCG, and RES can modulate various miRNA expressions, and these polyphenols play a pivotal role in ROS-mediated pathways and inflammation [151]. Further elucidation of the intricate mechanisms by which dietary polyphenols influence inflammation via MMPs and miRNA is necessary for advancing therapeutic strategies against inflammatory diseases. 

MDA has provided the structural basis for the interactions between polyphenols and MMPs to understand their MMP inhibitory activity and would be useful in exploring new polyphenol-based drugs with anti-inflammatory property. An example is the finding that MDA showed that in DenCUR, a CUR derivative, oleoyl, and polyethylene glycol moieties can contribute to the increase in two hydrogen bonds to interact with the enzyme, explaining the higher biological activity of DenCUR than CUR (Figure 6) [136]. A strategy to explore the modification of polyphenols that can increase MMP-interacting moieties would be useful in providing a better MMP inhibitor as an agent against inflammation-based human diseases.

Further advances in technology and research are expected to continue to unravel the detailed interaction between polyphenols and MMPs. The use of emerging computational technologies including quantum mechanical/molecular mechanical simulations [152,153], may enhance our understanding of precise binding modes and complexation between polyphenols and MMPs. 

## Figures and Tables

**Figure 1 molecules-28-05426-f001:**
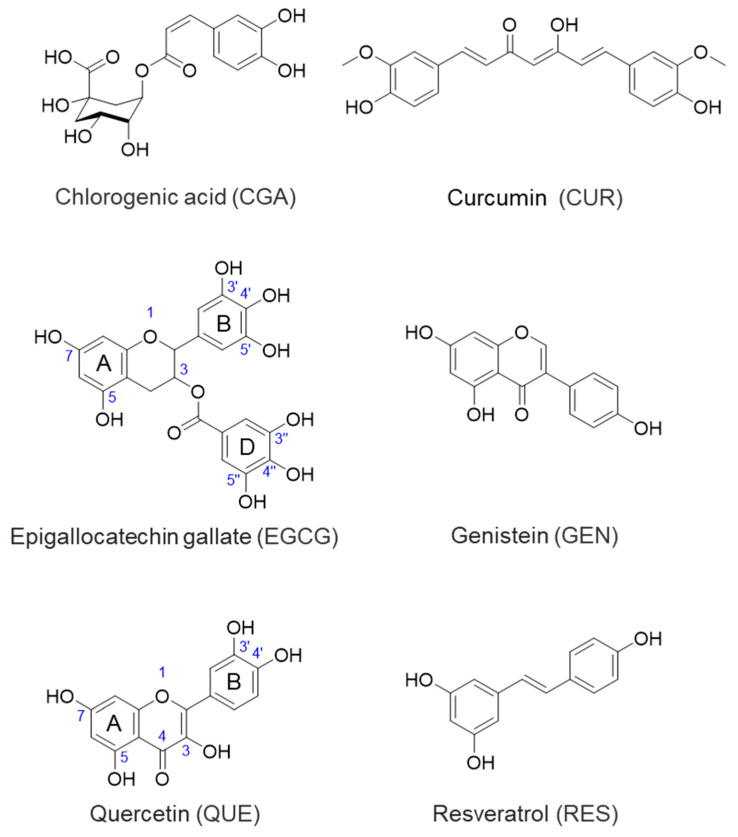
Chemical structure of dietary polyphenols. The numbering indicates the position of the carbon atom described in the main text.

**Figure 2 molecules-28-05426-f002:**
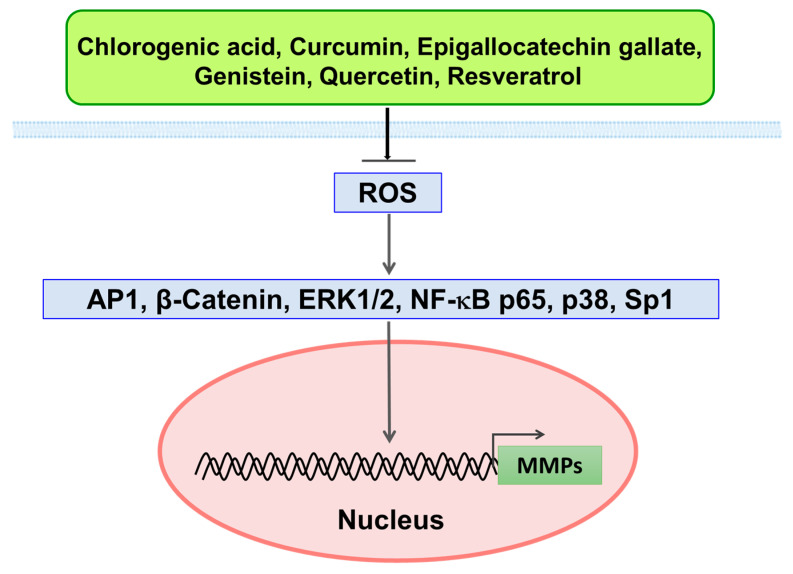
Downregulation of transcription factors by polyphenols through antioxidative action, leading to downregulations of MMPs.

**Figure 3 molecules-28-05426-f003:**
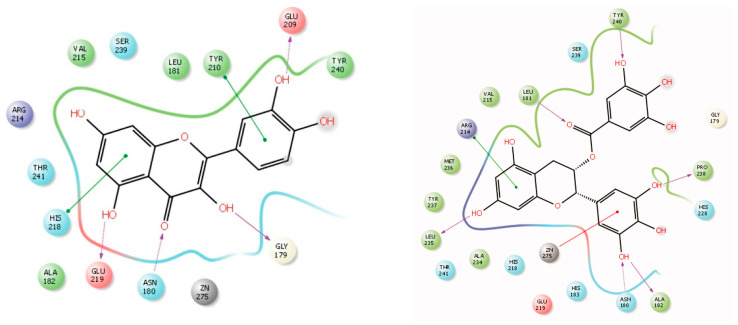
Two-dimensional diagrams of MMP1–polyphenol interaction for QUE (left) and EGCG (right) by MDA. The arrows indicate hydrogen bond between amino acid and each polyphenol. Reprinted with permission from [132]. Copyright 2020 Taylor & Francis.

**Figure 4 molecules-28-05426-f004:**
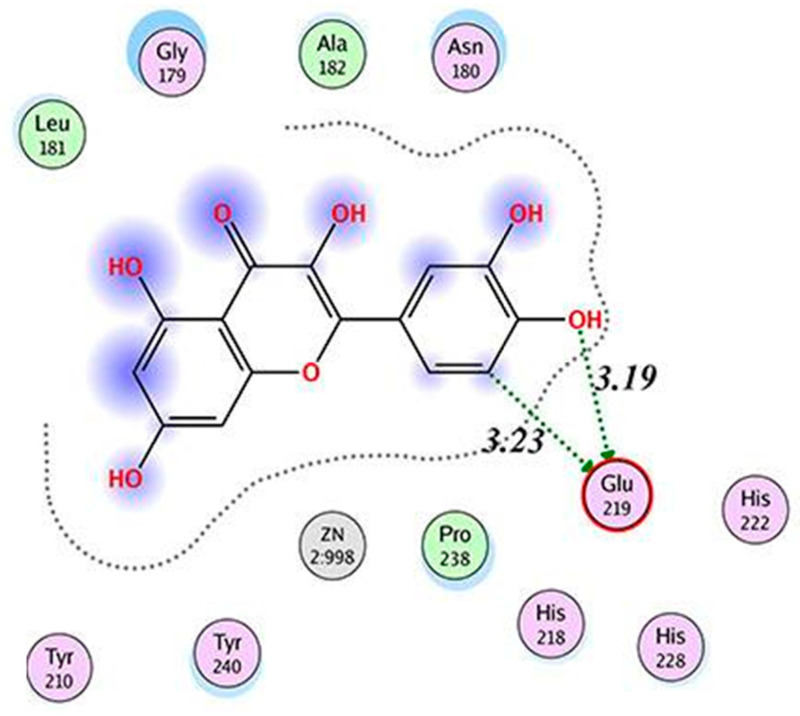
The model of interaction between QUE and MMP1. The interaction is indicated by the dotted arrows. Reprinted with permission from [134]. Copyright 2020 International Scientific Literature, Ltd.

**Figure 5 molecules-28-05426-f005:**
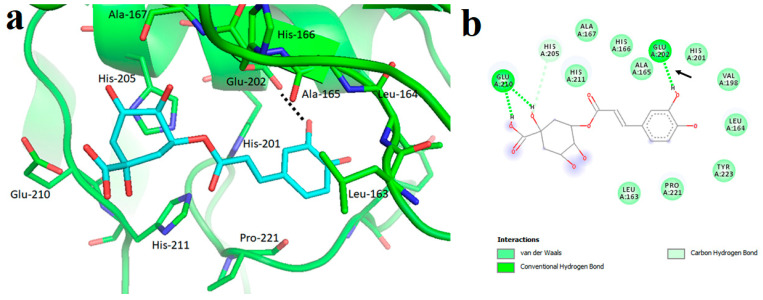
Interaction between MMP2 and CGA. (**a**) MDA of CGA (carbon atoms in blue) within the MMP2 active site. The interaction is indicated by the dotted line. (**b**) Two-dimensional interaction diagram between the CGA and MMP2. The hydrogen bond between the dihydroxyphenyl structure in CGA and Glu202 is shown by the arrow. Reprinted with permission from [135]. Copyright 2020 SciELO—Scientific Electronic Library Online.

**Figure 6 molecules-28-05426-f006:**
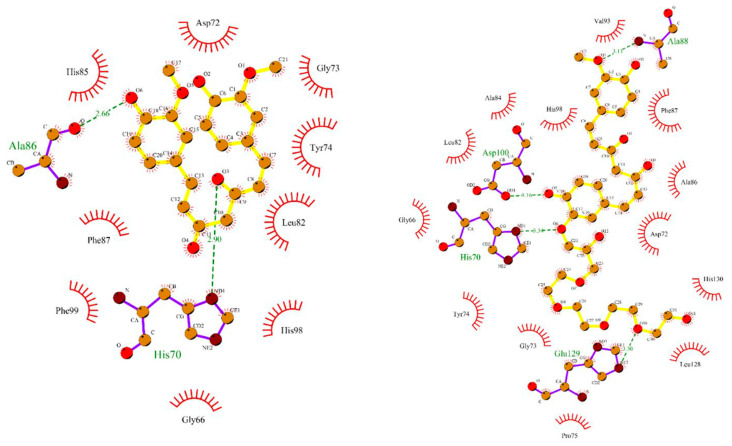
MDA of the interaction of MMP2 with CUR (left) and DenCUR (right). Ligands are yellow, amino acids involved in the complex are purple, hydrogen bonds are green dotted lines, and side residues are red. Reprinted with permission from [136]. Copyright 2023 BioMed Central.

**Figure 7 molecules-28-05426-f007:**
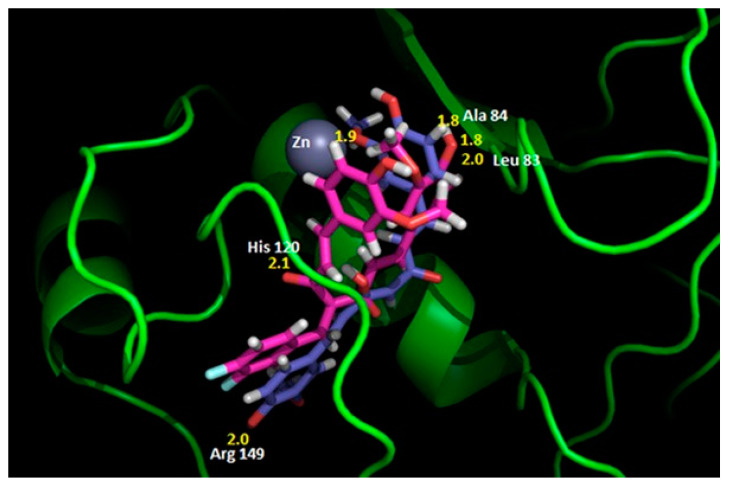
The superimposed image of the interaction of CUR (blue) or its difluorinated benzylidene derivative (cyan) with the MMP2 catalytic domain. Reprinted with permission from [138]. Copyright 2015 PubMed Central.

**Figure 8 molecules-28-05426-f008:**
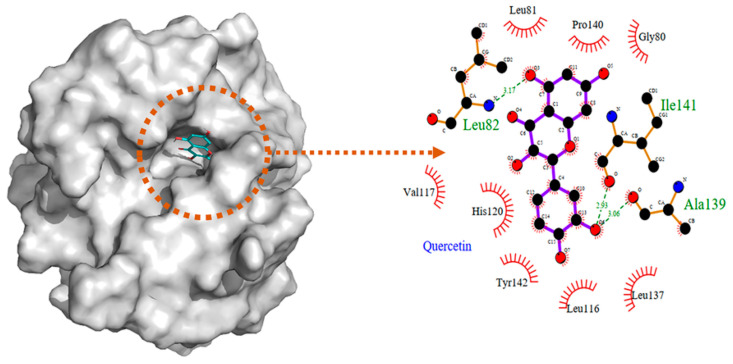
MDA of the interaction between QUE and MMP2. Reprinted with permission from [140]. Copyright 2022 PubMed Central.

**Figure 9 molecules-28-05426-f009:**
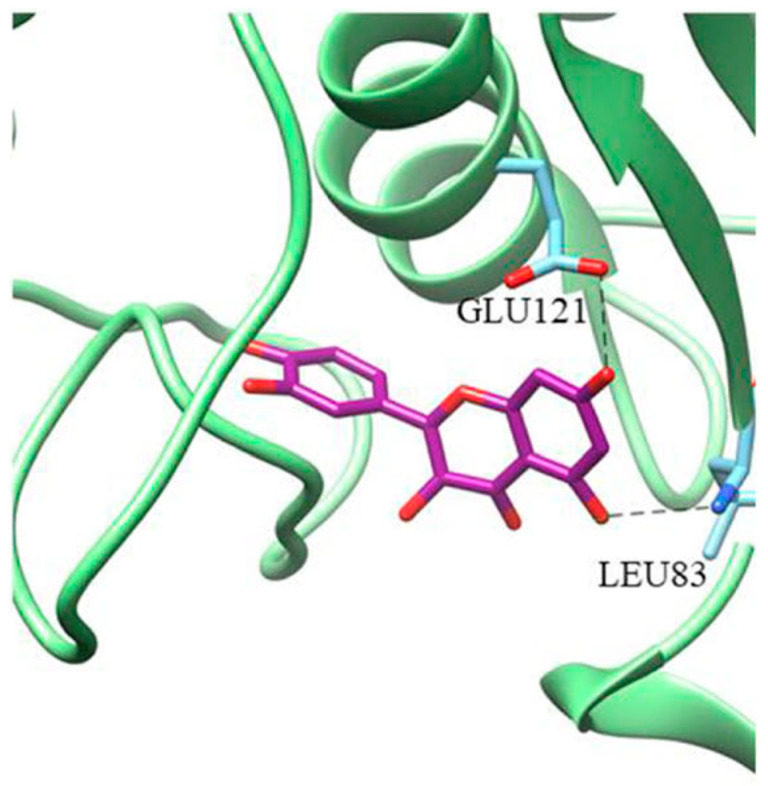
Three-dimensional interaction image of a stable complex of QUE with MMP2. Reprinted with permission from [141]. Copyright 2021 Springer Nature.

**Figure 10 molecules-28-05426-f010:**
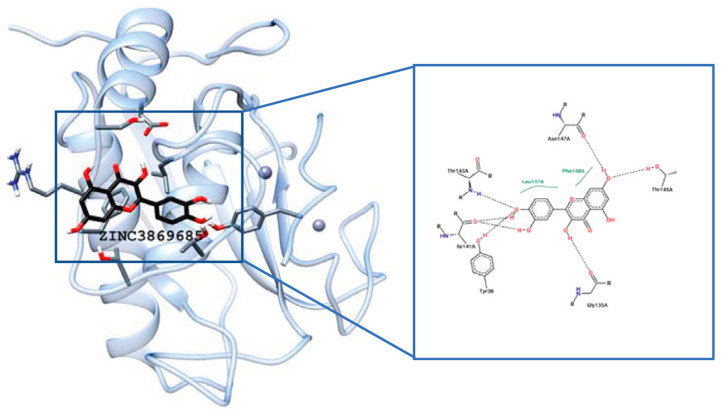
MDA of interaction between MMP2 and QUE. Reprinted with permission from [142]. Copyright 2021 Hindawi Limited.

**Figure 11 molecules-28-05426-f011:**
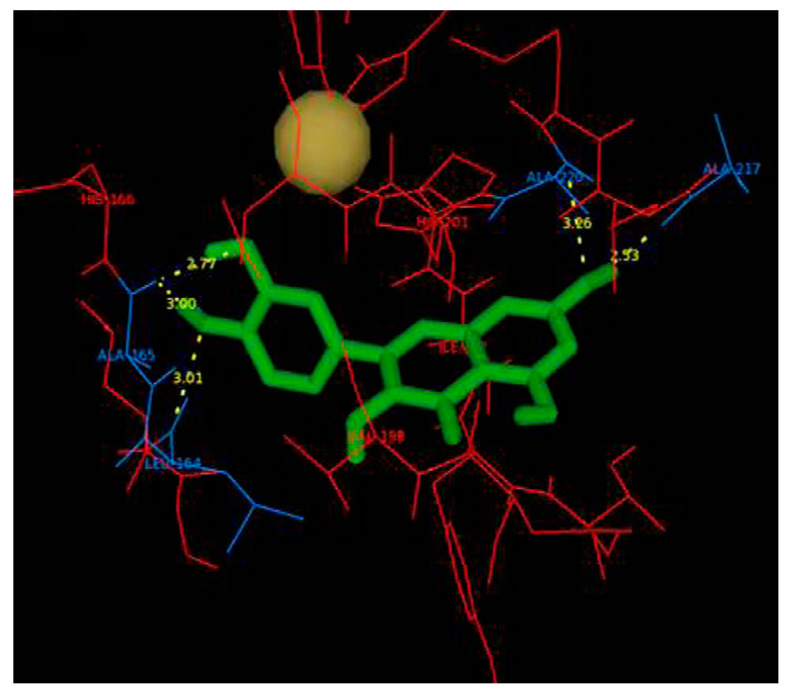
MDA of interaction between MMP2 and the QUE complex. Reprinted with permission from [143]. Copyright 2012 Elsevier Science.

**Figure 12 molecules-28-05426-f012:**
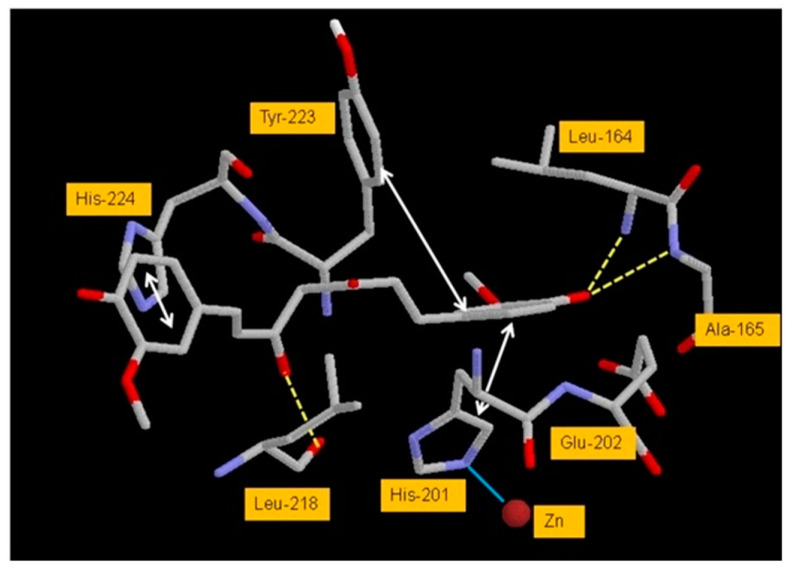
MDA of interaction between CUR and the active site residues of MMP3. Yellow dotted lines, cyan solid line, and double headed arrows represent hydrogen bonds, interaction with Zn, and π–π interactions, respectively. Reprinted with permission from [144]. Copyright 2015 PubMed Central.

**Figure 13 molecules-28-05426-f013:**
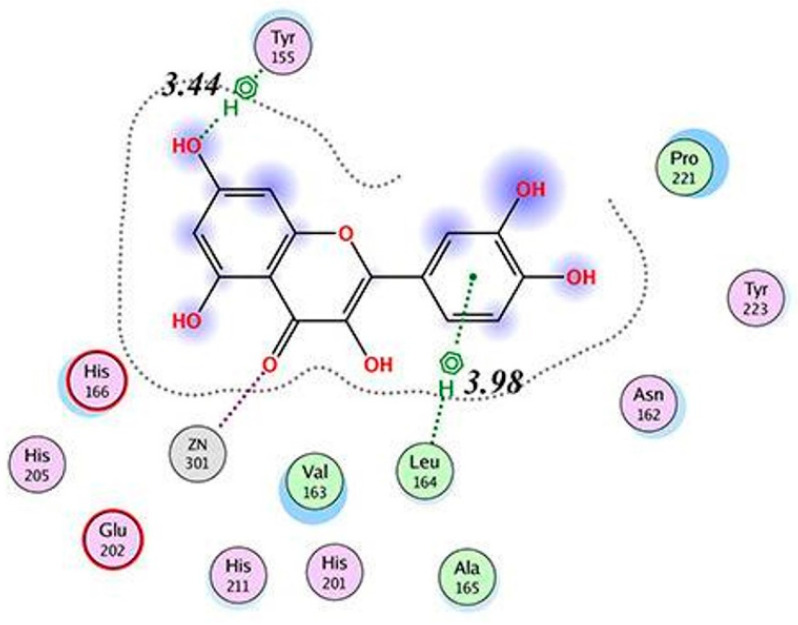
Two-dimensional presentation of the interaction between QUE and MMP3 referred to the report by Zhang et al. [134] [PMID 32866138]. Reprinted with permission from [134]. Copyright 2020 International Scientific Literature, Ltd.

**Figure 14 molecules-28-05426-f014:**
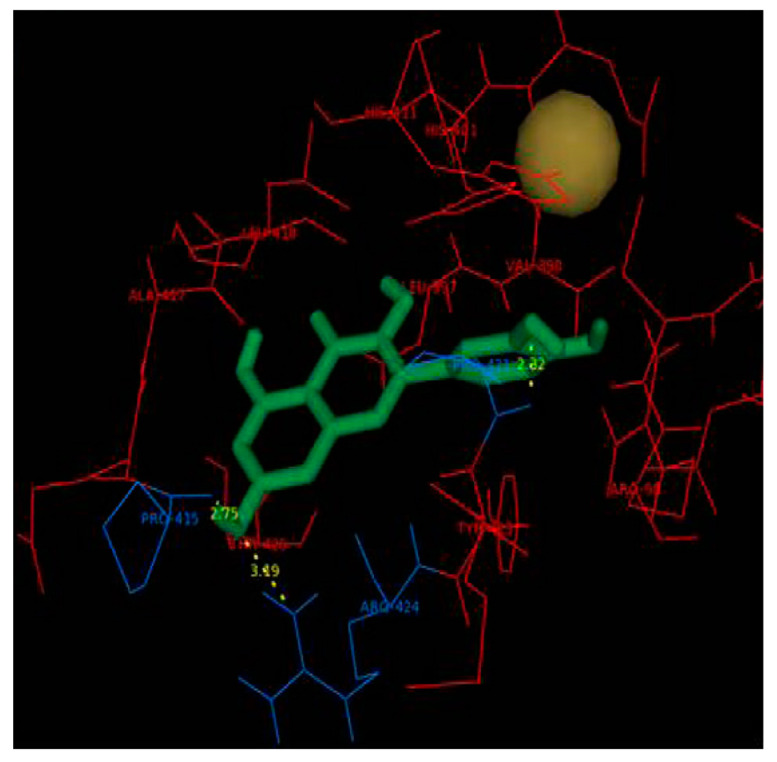
MDA of the interaction of the MMP9-QUE complex. Reprinted with permission from [143]. Copyright 2012 Elsevier Science.

**Figure 15 molecules-28-05426-f015:**
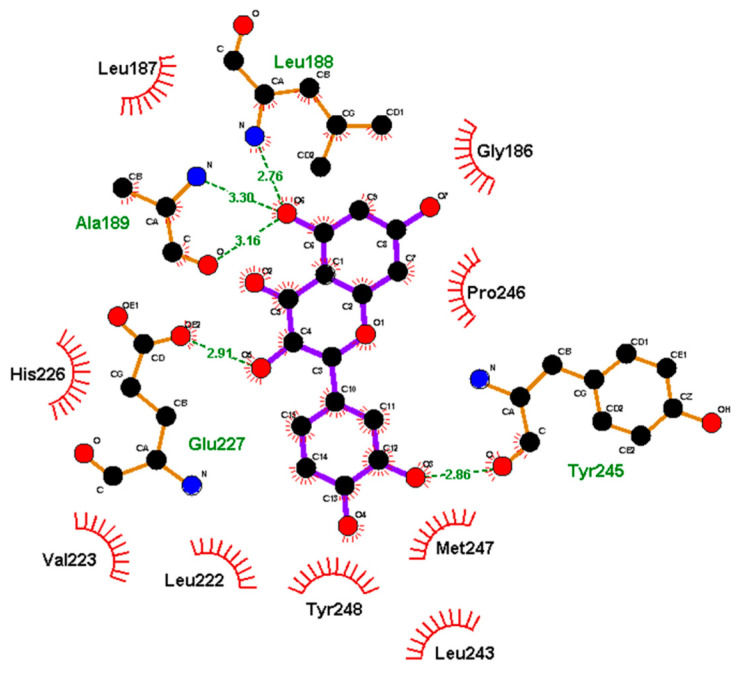
MDA for MMP9–QUE binding. Reprinted with permission from [147]. Copyright 2012 John Wiley & Sons Ltd.

**Figure 16 molecules-28-05426-f016:**
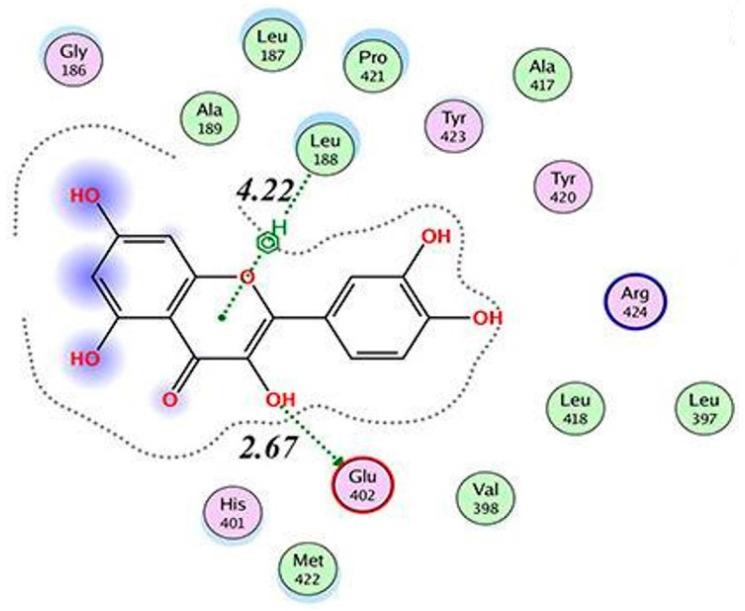
MDA for interaction of QUE with MMP9. Copyright 2022 International Scientific Literature, Ltd.

**Figure 17 molecules-28-05426-f017:**
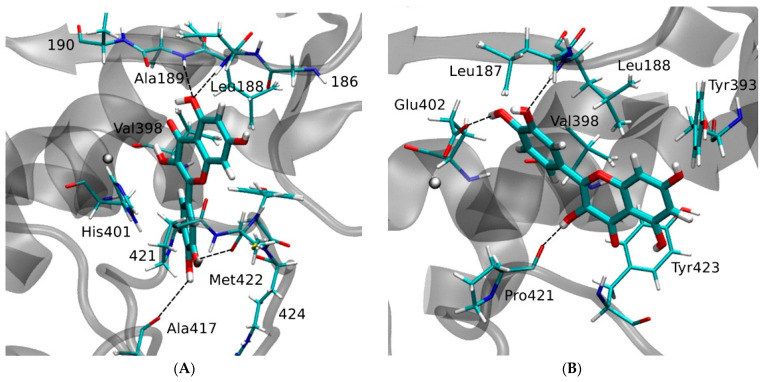
MDA for two representative models of the QUE-MMP9 complex, structure (**A**) (left) and (**B**) (right). Reprinted with permission from [148]. Copyright 2010 Elsevier Science.

**Table 2 molecules-28-05426-t002:** The literature showing downregulation of transcription factors involved in MMP expression.

	CGA	CUR	EGCG	GEN	QUR	RES
AP1	Cichocki et al. [101]	Sarawi et al. [102]	Fang et al. [103]	Wang et al. [104]	Hwang et al. [105]	Kim et al. [106]
β-catenin	Hu et al. [107]	Mohamadian et al. [108]	Fang et al. [103]	Yeh et al. [109]	Murata et al. [110]	Xie et al. [111]
ERK1/2	Gao et al. [112]	Mohamadian et al. [108]	Liang et al. [113]	Yeh et al. [109]	Ye et al. [114]	Chang et al. [115]
Hsp27	NF *	Tikoo et al. [116]	Yang et al. [117]	Xu et al. [118]	Sang et al. [119]	Díaz-Chávez et al. [120]
NF-κB	Moslehi et al. [121]	Sarawi et al. [102]	Fang et al. [103]	Nabavi et al. [122]	Cho et al. [123]	Kim et al. [106]
Specificity protein 1 (Sp1)	NF *	Liu et al. [124]	Fang et al. [103]	Miyamoto et al. [125]	Lee et al. [126]	Zeng et al. [127]
p38	Tan et al. [128]	Tikoo et al. [116]	Fechtner et al. [61]	Xu et al. [118]	Ye et al. [114]	Chang et al. [115]

* No relevant literature was found.

**Table 3 molecules-28-05426-t003:** Binding energy of polyphenols to MMPs and amino acid residues involved in hydrogen bonding.

	QUE	CUR	EGCG
MMP1	−7.15 kcal/mol ^(1)^ (Glu219)		
MMP2	−8.17 kcal/mol ^(2)^ (Leu82, Ala139, Ile141) −10.1 kcal/mol ^(3)^ (Leu83, Val117, Glu121, Ala136) −7.90 kcal/mol ^(4)^ (Tyr3, Ile141, Thr143, Thr145, Asn147, Phe148) −9.11 kcal/mol ^(5)^ (Leu164, Ala165, Ala217, Ala220)	−7.35 kcal/mol ^(8)^ (Arg149)	−32.72 kcal/mol ^(10)^ (Leu399, His403, Glu404, Ala192)
MMP3	−7.25 kcal/mol ^(1)^ (Tyr155, Leu164)	−10.2 kcal/mol ^(9)^ (Leu164, Ala165, Leu218)	
MMP9	−8.82 kcal/mol ^(5)^ (Pro415)−10.8 kcal/mol ^(6)^ (Leu188, Ala189, Glu227, Tyr245) −6.16 kcal/mol ^(1)^ (Glu402)−9.9 kcal/mol ^(7)^ (Leu188, Ala189, Glu227, Met247)		
MMP14			−57.61 kcal/mol ^(10)^ (Leu199, Phe234, His239, Glu240, Met257, Gln262)

Amino acid residues involved in hydrogen bonding are indicated in parentheses. The binding energies were taken from the following references: ^(1)^ Zhang, Z. et al. [134], ^(2)^ Li, F. et al. [140], ^(3)^ Erusappan, T. et al. [141], ^(4)^ Xu, J. et al. [142], ^(5)^ Pandey, A.K. et al. [143], ^(6)^ Yu, J. et al. [147], ^(7)^ Huynh, C.B. et al. [149], ^(8)^ Ahmad, A. et al. [138], ^(9)^ Jerah, A. et al. [144], ^(10)^ Chowdhury, A. et al. [139].

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
