# Peer review of "Anti-Inflammatory Effects of Dietary Polyphenols through Inhibitory Activity against Metalloproteinases"

_molecules, 2023, doi:10.3390/molecules28145426_

Round 1
Reviewer 1 Report
This is an interesting piece of manuscript that reports novel findings and can be accepted for publication after addressing these minor concerns.
- Dietary polyphenols have been demonstrated to have a variety of beneficial effects on human diseases such as cancer, obesity, diabetes, cardiovascular disease, and neurodegenerative disorders. Some recent studies reported many important effects of polyphenols that can be included here. https://doi.org/10.1016/j.ijbiomac.2022.03.004; https://doi.org/10.1007/978-981-16-4558-7_7; https://doi.org/10.3390/nu15071704; https://doi.org/10.1155/2020/1245875
- Maintain the consistent usage of abbreviations throughout the manuscript.
- Figure 2…Why use short forms in figures? Avoid usage of abbreviations.
- Improve Figure 3 quality. No residues can be seen.
- Figure 6 needs to be improved as everything appears blurry.
- Future perspectives need to be added.
- Conclusion needs to be improved.
Author Response
Thank you very much for your precious and constructive comments. We have responded as attached Word file.

Reviewer 2 Report
1. The article uses figure3 as an example, the picture in the article is too blurry, please provide a clear enough picture of the results.
2. In Figure1, some of the molecular formulae of polyphenols are identified by numbers, please explain what the numbers represent.
3. It is recommended that the authors add a summary table of binding energies and hydrogen bonding of several polyphenols to MMPs to the summary. 3.
4. Suggest that the authors add an analysis of the structure and activity results for polyphenols.
5. Please check the spelling of words in the article for correctness.
6. Is there any other computer simulation analysis of the relationship between polyphenols and MMPs and inflammation, such as quantum mechanics, in addition to computational molecular docking analysis.
7. Are the 6 polyphenols mentioned in this article the most widely studied and best active dietary polyphenols and why were these chosen?
Author Response

(The authors gave the same response as above.)
